# Perioperative anticoagulation in patients with intracranial meningioma: No increased risk of intracranial hemorrhage?

**Florian Wilhelmy**[1☯]*, **Annika Hantsche**[1☯], **Tim Wende**[1], **Johannes Kasper**[1], **Vera Reuschel**[2], **Clara Frydrychowicz**[3], **Stefan Rasche**[4], **Dirk Lindner**[1], **Jürgen Meixensberger**[1]

1 Department of Neurosurgery, University Hospital Leipzig, Leipzig, Germany, 2 Division of Neuroradiology, University Hospital Leipzig, Leipzig, Germany, 3 Division of Neuropathology, University Hospital Leipzig, Leipzig, Germany, 4 Department of Anesthesiology and Intensive Care, University Hospital Leipzig, Leipzig, Germany

☯ These authors contributed equally to this work.
* florian.wilhelmy@medizin.uni-leipzig.de

## Abstract

### Objective

Anticoagulation (AC) is a critical topic in perioperative and post-bleeding management. Nevertheless, there is a lack of data about the safe, judicious use of prophylactic and therapeutic anticoagulation with regard to risk factors and the cause and modality of brain tissue damage as well as unfavorable outcomes such as postoperative hemorrhage (PH) and thrombo-embolic events (TE) in neurosurgical patients. We therefore present retrospective data on perioperative anticoagulation in meningioma surgery.

### Methods

Data of 286 patients undergoing meningioma surgery between 2006 and 2018 were analyzed. We followed up on anticoagulation management, doses and time points of first application, laboratory values, and adverse events such as PH and TE. Pre-existing medication and hemostatic conditions were evaluated. The time course of patients was measured as overall survival, readmission within 30 days after surgery, as well as Glasgow Outcome Scale (GOS) and modified Rankin Scale (mRS). Statistical analysis was performed using multivariate regression.

### Results

We carried out AC with Fraxiparin and, starting in 2015, Tinzaparin in weight-adapted recommended prophylactic doses. Delayed (216 ± 228h) AC was associated with a significantly increased rate of TE (p = 0.026). Early (29 ± 21.9h) prophylactic AC, on the other hand, did not increase the risk of PH. We identified additional risk factors for PH, such as blood pressure maxima, steroid treatment, and increased white blood cell count. Patients'

**Data Availability Statement:** All relevant data are within the manuscript and its Supporting Information files.

**Funding:** We acknowledge support from Leipzig University for Open Access Publishing.

**Competing interests:** The authors have declared that no competing interests exist.

**Abbreviations:** AC, anticoagulation; BMI, Body Mass Index; NO, no event group; CT/CAT scan, computed axial tomography scan; CI, confidence interval; diast., diastolic; DOAC, direct oral anticoagulant; DVT, deep vein thrombosis; F, female; GOS, Glasgow Outcome Scale; ICD-10, International Classification of Diseases (10th Revision); INR, international normalized ratio; IPC, intermittent pneumatic compression; IQR, interquartile range; IS, ischemic stroke; LMWH, low-molecular-weight heparin; M, male; mRS, modified Rankin Scale; NOAC, non-vitamin K antagonist oral anticoagulants; n.s., not significant; OR, odds ratio; PE, pulmonic embolism; PH, postoperative hemorrhage; PHR, postoperative hemorrhage–revised; pTT, partial thromboplastin time; s., see; syst., systolic; TE, thromboembolic event; t-test, Welch's t-test; UFH, unfractionated heparin; U test, Mann–Whitney U test; VIF, variance inflation factor; WBC, white blood cell.

outcome was affected more adversely by TE than PH (+3 points in modified Rankin Scale in TE vs. +1 point in PH, p = 0.001).

## Conclusion

Early prophylactic AC is not associated with an increased rate of PH. The risks of TE seem to outweigh those of PH. Early postoperative prophylactic AC in patients undergoing intra-cranial meningioma resection should be considered.

## Introduction

### Clinical and scientific background

In the perioperative treatment of patients undergoing intracranial surgery, we are caught between two stools: the need for prophylactic AC and its most feared adverse effect, intracranial hemorrhage. As the latter is always a life-threatening event, with a mortality rate between 18 and 32% [1, 2], neurosurgeons naturally tend to avoid perioperative AC as much as possible [2, 3]. Neurosurgical patients are at high risk of thromboembolism due to immobilization and the long duration of surgery. Additionally, with cardiovascular diseases becoming more prevalent owing to demographic changes, increasing numbers of significantly anticoagulated patients are to be observed [4]. A survey conducted by Skardelly et al. concluded that in 2016, nearly two thirds of German neurosurgical departments had not yet defined an algorithm for continuous, discontinuous or bridged anticoagulant and antiplatelet therapy during elective surgery. Therefore, treatment standards for patients at individual risk of TE vary widely among different centers. In some cases, altering, pausing, or stopping their blood thinning regimen prior to intracranial surgery puts them at high risk of TE [5].

Owing to the lack of sound data, the decision regarding the type, dose and time point for perioperative prophylactic anticoagulation is frequently based on clinical experience, estimated risk, and interdisciplinary discussion between cardiology, intensive care practitioners, and neurosurgeons. Relevant indications (e.g. aortic valve replacement, atrial fibrillation, stents), risk factors for TE (immobilization, prolonged duration of surgery, prone position during surgery, coagulopathy, malignant diseases, infection, comorbidities, history of TE, old age, hemiparesis or hemiplegia) [2, 3, 6–12] are weighed against risk factors for PH (intraoperative bleeding tendency, size and vascularization of the tumor, entity, prolonged operative time, metabolic syndrome). The variety of risk factors and intra-individual differences regarding tumor morphology, AC, and corresponding indication is vast [1, 2, 6–8, 13]. Therefore, our study focused on the search for independent yet widespread, easily measurable risk factors for PH and TE in patients with cranial meningiomas.

### Objective and outlook

In this paper, we present retrospective data on perioperative anticoagulation in meningioma surgery. We aim to improve safe perioperative prophylactic AC management and contribute to guidelines and standards.

## Patients and methods

### Patient selection and treatment

We searched the digital database of University Hospital Leipzig for all patients diagnosed with meningioma who had undergone surgery between 2012 and 2018 and been operated on at the

Department of Neurosurgery, University Hospital Leipzig. We included 286 patients with no exclusions. Data acquisition was approved by the ethics committee of the Medical Faculty, University of Leipzig (No. 053/19-ek). All patient data were fully anonymized. Being an anonymous retrospective review with no personal data, the ethics committee did not require informed consent.

Exclusion criteria included missing documentation or inconclusive data on the perioperative AC regimen as well as age below 18 years. Patients lost to or without follow-up within 30 days were excluded for acute treatment evaluation.

All tumor cases in our neurooncological center were discussed in a weekly, interdisciplinary tumor board. The therapy regimen regarding operative resection or conservative treatment was decided in interdisciplinary discourse based on international guidelines [EANO].

Patients were anticoagulated in accordance with national guidelines [14, 15], known independent risk factors, and pre-existing diseases. The time point and dose as well as the choice of substance were decided according to the hospital's guidelines and in interdisciplinary bedside discourse with intensive care and neurosurgical practitioners. The fact that anticoagulation did not follow a definite protocol resulted in the variety of time points and therapy regimens examined in this study. However, the common course of treatment and diagnosis can be described as follows.

Patients are not anticoagulated until a CT scan has been performed. CT scans on the first postoperative day are routine (within 24h after surgery). CT scans are performed immediately following the onset of new neurological deficits.

We begin prophylactic AC treatment in patients undergoing intracranial surgery on the first day after surgery, as long as postoperative CT scans do not show any signs of residual bleeding. Patients with postoperative or perioperative hemorrhage are mainly anticoagulated after 3–5 days following a repeat CT scan without any sign of a growing hemorrhage.

In patients with a preoperative regimen of DOAC or NOAC, therapy was interrupted for up to 3 weeks depending on indications during the period concerned. In these cases, heparin bridging was regularly performed.

Since 2014, a standard operating procedure has stated that hospitalized tumor patients should be considered for anticoagulation as long as there are no contraindications, in which case either mechanical prophylaxis or no prophylaxis at all should be applied. We have adapted this regimen and currently halt pre-existing AC until the removal of sutures if there are no urgent indications.

## Assessed data

Laboratory values were assessed at the time of admission, as were minima and maxima. PH was assessed as an adverse event (0 = did not occur, 1 = occurred), and also classified regarding the need for surgical intervention. TE (i.e. mesenterial emboli, pulmonary emboli, deep vein thrombosis, cerebral or myocardial infarction) were classified accordingly (0 = did not occur, 1 = occurred). We distinguished between the localization of embolism for further analysis.

Selected parameters:

Biographical: Age, gender, weight, height, smoker, pre-existing anticoagulation (indication and regimen).

Laboratory charts: pTT, INR, blood pressure, platelet count, hematocrit, creatinine, sodium, white blood cell count, indicators for coagulopathy.

Treatment: Postoperative anticoagulation, time point of treatment initiation, prevalence of hemorrhage and embolism, regimen (dose/substance/application). Antiplatelet therapy was evaluated, but temporarily paused in all cases.

Peri-operative: General anesthesia, steroid treatment, dialysis, duration of surgery, type of procedure/bleeding, GOS/mRS after surgery.

End points: Postoperative hemorrhage (CT morphological) with or without revision, pulmonary emboli, thrombosis (vein or peripheral), death, readmission (30 days).

## Statistical analysis

Continuous parameters are displayed as median with standard deviation and interquartile range and were analyzed with Welch's Test, t-test and Mann–Whitney U test (although only the Mann–Whitney U test was used for GOS and mRS as ordinal variables).

Dichotomous parameters are shown as number and percentage, and statistically compared using chi-square or Fisher's exact tests.

P-values lower than 0.05 were considered statistically significant.

Odds ratios (OR) and their confidence interval of 95% were computed using univariate binomial logistic regression. Parameters which appeared relevant for the occurrence of TE or PH in univariate analysis were included in multiple logistic regression. We attempted to find the model with the best fit. Collinearity between the independent variables was tested using the variance inflation factor (VIF). The Hosmer–Lemeshow test was applied to confirm calibration.

All analyses were completed using IBM SPSS Statistics Version 24 (IBM, Armonk, New York State, USA).

## Results

A total of 286 patients with intracranial meningioma who had been operated on in the Department of Neurosurgery between 2012 and 2018 were included in this study. The 3-month follow-up data for patient-reported adverse events were scanned. Patient characteristics and parameters are summarized in Tables 1 and 2. Intracranial residual blood, radiologically detected by 5 mm and 1.25 mm CT scans, was taken into account. Moreover, patients were divided into two groups: those suffering from postoperative hemorrhage (PH) and those with hemorrhage necessitating re-operation (PHR). The following parameters were identified as correlated to a) TE or b) PH (Tables 1 and 2):

## Epidemiology

286 patients with meningioma who had undergone surgery were analyzed. 11 (3.8%) patients developed thromboembolic events (TE): 7 patients with pulmonary embolism (PE, 2.4%), 6 with deep vein thrombosis (DVT, 2.1%). 2 patients suffered from ischemic stroke (IS, 0.7%). 17 patients (5.9%) suffered from postoperative intracranial hemorrhage (PH), as seen in the mandatory postoperative CT scan or in the event of postoperative neurological deficit, of whom 9 (3.1%) had to be surgically revised (PHR).

There was no significant difference in epidemiological characteristics such as age, BMI or gender. Higher age trended to be correlated with TE, albeit not significantly (p = 0.053).

## Time point of prophylactic anticoagulation

Patients with delayed prophylactic AC (point of treatment in hours after surgery) were more likely to suffer from TE (NO 29 ± 21.9h, TE 54 ± 69.2, p = 0.301). The OR showed an increased risk for patients with delayed prophylactic anticoagulation of suffering from TE (OR 1.027, CI 1.010–1.043). In the subgroup analysis, 2 patients were found to have suffered from TE immediately after surgery (PE after 2h, DVT after 24h), which changed their course of treatment

**Table 1. Demographic description and risk factors for thromboembolic events (TE) in operated meningioma patients.**

| Characteristic | NO (n = 275) | TE (n = 11*) | p-value ** |
|---|---|---|---|
| **Demographic** | | | |
| **Gender** | 82 M (29.8%) | 3 M (27.3%) | 1.000 |
| | **193 F (70.2%)** | 8 F (72.7%) | |
| **Age** | 61 ± 13 (50–72) | 71 ± 12 (56–79) | 0.059 |
| **BMI** | 26 ±5 (24–29) | 29 ± 4 (26–34) | 0.059 |
| **Medical history** | | | |
| **Pre-existing anticoagulation** | 39 (14.2%) | 2 (18.2%) | 0.769 |
| **Smoker** | 34 (12.4%) | 3 (27.3%) | 0.164 |
| **Steroid medication** | 145 (52.7%) | 7 (63.6%) | 0.493 |
| **Coagulation disorder** | 12 (4.4%) | 2 (18.2%) | 0.095 |
| **Laboratory values** | | | |
| **Minimum platelet count** | 178 ± 59.1 (140–216) | 139 ± 32.7 (107–150) | 0.001 |
| **Minimum hematocrit** | 0.301 ± 0.050 (0.273–0.335) | 0.223 ± 0.061 (0.206–0.292) | 0.006 |
| **Minimum INR** | 1.1 ± 5.3 (1.2–1.05) | 1.2 ± 5.3 (1.4–1.1) | 0.001 |
| **Maximum blood pressure** | 155/65 ± 22/14 (140/60–170/75) | 175/80 ± 20/13 (155/70–190/85) | Syst: 0.005 Diast: 0.016 |
| **White blood cell count on admission** | 7.6 ± 3.4 (6.1–9.7) | 8.9 ± 6.1 (7.6–15.8) | 0.025 |
| **Maximum white blood cell count** | 13 ± 6.0 (10–16.7) | 16.6 ± 4.6 (15.6–20.5) | 0.01 |
| **Event** | | | |
| **Length of surgery** | 269 ± 123.3 (196–356) | 373 ± 137.8 (257–518) | 0.015 |
| **Time point of anticoagulation** | 29 ± 21.9 (27–51) | 54 ± 69.2 (24–96) | 0.301 |
| **Time point of anticoagulation, revised***** | 29 ± 21.9 (27–51) | 216± 228 (62–400) | 0.026 |

*2 × DVT, 3 × PE, 4 × DVT+PE, 2 × ischemic stroke

**Metric parameters: U test, ordinal parameters: chi-square test

***2 patients were excluded for additional statistical analysis due to immediate TE (see below).

(immediate therapeutic treatment with low-molecular-weight heparin (LMWH, Tables 3 and 4). We took this significant change in treatment from prophylactic to therapeutic AC into account by excluding these patients and performing additional subgroup analysis regarding the time point of AC and thromboembolic risk. There is, of course, no change in the overall risk of TE after said adjustment; late prophylactic anticoagulation was significantly correlated with thromboembolic events (p = 0.026). The OR was 1.035 (CI 1.015–1.055). Two cases of intracranial hemorrhage were discovered before the administration of anticoagulation (18h and 20h postoperative), which was therefore paused.

Early AC did not lead to an increased rate of postoperative hemorrhage. By contrast, AC was delayed in patients with PH (in h after surgery: GC 29 ± 22.4, PH 69 ± 54 p = 0.0007; PHR 28 ± 73.7; p = 0.572). OR showed decreased risk of TE and PH in early anticoagulated patients (TE 0.974, CI 0.958–0.990, PH 0.972, CI 0.957–0.988, PHR 0.977 CI 0.961–0.993).

Overall, there were significant differences in the time point of administration.

81.25% of postoperative hemorrhages occurred within 24 hours after surgery. In only 18.75% of cases did bleeding appear on the CT scan after anticoagulation had been administered.

By contrast, only 36.4% of the TE group were anticoagulated within 24 hours after surgery, and of those, 18.2% were given heparin in therapeutic doses.

Patients with coagulation disorder showed increased risk of PH (PH 17.6%, p = 0.042; PHR 33.3%, p = 0.0007).

**Table 2. Demographic description and risk factors for postoperative hemorrhage (PH) in operated meningioma patients.**

| Characteristic | NO (n = 269) | PH (n = 17) | p value* |
|---|---|---|---|
| **Demographic** | | | |
| **Gender** | 78 M (29%) | 7 M (41.2%) | 0.287 |
| | **191 F (71%)** | 10 F (58.8%) | |
| **Age** | 61 ± 13 (50–72) | 67 ± 14 (49–77) | 0.573 |
| **BMI** | 27 ±5 (24–30) | 24 ± 3 (24–28) | 0.735 |
| **Medical history** | | | |
| **Pre-existing anticoagulation** | 37 (13.8%) | 4 (ASS, 23.5%) | 0.414 |
| **Smoker** | 36 (13.4%) | 1 (5.9%) | 0.709 |
| **Steroid medication** | 136 (50.6%) | 16 (94.1%) | 0.001 |
| **Coagulation disorder** | 11 (4.1%) | 3 (17.6%) | 0.042 |
| **Event** | | | |
| **Length of surgery** | 270 ± 122 (192–358) | 305 ± 156 (232–454) | 0.130 |
| **Time point of anticoagulation** | 29 ± 22.4 (27–50) | 69 ± 54 (28–96) | 0.007 |
| **Time point of event (after surgery) in h** | | 19 ± 16 (15–23) | |
| **Laboratory values** | | | |
| **Maximum platelet count** | 267 ± 91.6 (222–321.5) | 335 ± 129.4 (256.5–452) | 0.013 |
| **Minimum platelet count** | 177 ± 57.6 (140–215.5) | 142 ± 81.0 (119–201.5) | 0.106** |
| **Minimum hematocrit** | 0.301 ± 0.050 (0.273–0.335) | 0.246 ± 0.060 (0.204–0.324) | 0.011 |
| **Maximum blood pressure** | 155/65 ± 22/14 (140/60–170/75) | 175/80 ± 19/12 (155/60–188/85) | Syst: 0.003 Diast: 0.053 |
| **White blood cell count on admission** | 7.5 ± 3.6 (6.2–9.8) | 9.1 ± 3.2 (8.0–11.6) | 0.045 |
| **Maximum white blood cell count** | 13 ± 5.6 (10–16.5) | 17.6 ± 8.8 (13.2–24.0) | 0.003 |
| **Creatinine on admission** | 70 ± 23.2 (63–81.5) | 85 ± 15.4 (67.5–88.5) | 0.016 |
| **Maximum creatinine** | 72 ±27.9 (64–85.5) | 90.0 ± 21.8 (76.5–115.5) | 0.001 |
| **Minimum creatinine** | 55 ±14.1 (48–65.5) | 52 ± 13.4 (45–65.5) | 0.553* |

*Metric parameters: U test, ordinal parameters: chi-square test

**Subgroup analysis of patients with surgically revised hemorrhage (PHR) was performed. In this group, a significant correlation was found for said hemorrhage with minimum platelet count (NO 175 ± 59.2 vs. PHR 120 ± 93.6, p = 0.007, OR 0.989 (CI 0.977–1.002)) as well as minimum creatinine (NO 55 ±14 vs. PHR 45 ± 7.2, p = 0.006, OR 0.939 (CI 0.892–0.990)). The complete subgroup analysis is contained in the appendix.

**Table 3. Patients with TE and time points of anticoagulation.**

| Time point of TE (h after procedure) | Time point of first AC (h after procedure) | Duration of procedure | Type of AC | Type of TE | PH/PHR |
|---|---|---|---|---|---|
| 2 | 3 | 338 | Therapeutic heparin (900 U/h) | PE | |
| 24 | 24 | 396 | Therapeutic heparin (1200 U/h) | DVT | |
| 62 | 23 | 591 | Prophylactic LMWH | PE | |
| 168 | 96 | 257 | Prophylactic LMWH | Ischemic stroke | Yes, 18h after procedure |
| 181 | 24 | 518 | Prophylactic LMWH | PE and DVT | |
| 216 | 72 | 583 | Prophylactic LMWH | PE and DVT | |
| 264 | 54 | 245 | Prophylactic LMWH | Ischemic stroke | |
| 400 | 240 | 169 | Prophylactic LMWH | PE and DVT (after discharge) | Yes, 20h after procedure |
| 400 | 48 | 453 | Prophylactic LMWH | DVT (after discharge) | |
| 567 | 149 | 355 | Prophylactic LMWH | PE | |
| 720 | 96 | | Prophylactic LMWH | PE and DVT | |

**Table 4. Patients with PH(R) and time point of anticoagulation.**

| Time of PH (CT diagnosis, h postoperative) | Time of first AC (in h postoperative) | Duration of procedure (min) | Surgically revised | Type of AC | TE |
|---|---|---|---|---|---|
| 4 | 98 | 448 | No | LMWH | No |
| 4 | 78 | 233 | No | LMWH | No |
| 4 | 73 | 461 | Yes | LMWH | No |
| 11 | 69 | 753 | Yes | LMWH | No |
| 18 | 96 | 257 | No | LMWH | ischemic stroke (168h postoperative) |
| 18 | 72 | 305 | No | LMWH | No |
| 18 | 52 | 231 | No | LMWH | No |
| 18 | 51 | 329 | No | LMWH | No |
| 19 | 53 | 224 | No | LMWH | No |
| 19 | None | 335 | No | None | No |
| 20 | 240 | 169 | Yes | LMWH | DVT and LAE (after discharge) |
| 22 | 112 | 375 | Yes | UFH | No |
| 22 | None | 255 | Yes | None | No |
| 24 | 28 | 216 | Yes | LMWH | No |
| 40 | 26 | 504 | Yes | LMWH | No |
| 53 | 24 | 584 | Yes | LMWH | No |
| 64 | 26 | 250 | Yes | LMWH | No |

## Multiple logistic regression

For the TE group, we found the following parameters to create the best fit: time point of prophylactic anticoagulation, maximum white blood cell count, white blood cell count on admission, minimum hematocrit, maximum systolic blood pressure, and minimum Quick value (Hosmer–Lemeshow chi-square value = 7.328, p = 0.502). Within this model, the following parameters remained statistically significant: time point of anticoagulation (p = 0.008, OR = 1.029, 95% CI 1.007–1.051), maximum white blood cell count (p = 0.049, OR = 0.834, 95%CI 0.695–0.999), white blood cell count (p = 0.028, OR = 1.246, 95% CI 1.024–1.515), minimum hematocrit (p = 0.021, OR = 0.0001, 95%CI 0.000–0.062).

Three predictors entered the PH model: time point of AC, maximum systolic blood pressure, and steroid usage (Hosmer–Lemeshow chi-square value = 12.415, p = 0.134). However, the only significant variable was the time point of AC (p = 0.042, OR = 1.016, 95%CI 1.001–1.032).

Detailed information on both models can be found in Appendix B.

## Outcome

Patients with post-operative TE as well as PH have a worse clinical outcome when analyzed for both GOS (GC 5 ± 0.7 vs. TE 3 ± 1.2 (p = 0.000) vs. PH 4 ± 1.3 (p = 0.002)) and mRS (NO 1 ± 1.2 vs. TE 4 ± 1.9 (p = 0.000) vs. PH 2 ± 1.8 (p = 0.001)). Moreover, TE are associated with a higher risk of reduced functional status than PH (median GOS 3 vs 4, median mRS 4 vs. 2). In-hospital mortality is higher in both subgroups than the no-event group (NO: 2 (0.8%); TE: 2 (18.2%), p = 0.013; PH: 2 (11.8%), p = 0.043) (see Table 5).

In total, 4 patients died on first admission. Two suffered from pulmonary embolism, one from uroseptic shock in an advanced state of kidney carcinoma, and one from ischemic stroke of unknown origin. No patient died solely from intracranial hemorrhage.

**Table 5. Differences in outcome parameters for patients suffering from TE and PH.**

| Outcome | NO (n = 275/269) | TE (n = 11) | | PH (n = 17) | |
|---|---|---|---|---|---|
| GOS (IQR) | 5 ± 0.7 (4–5) | 3 ± 1.2 (3–4) | 0.0001 | 4 ± 1.3 (3–5) | 0.002 |
| mRS (IQR) | 1 ± 1.2 (0–2) | 4 ± 1.9 (2–5) | 0.0001 | 2 ± 1.8 (1–4) | 0.001 |
| In-hospital mortality | 2 (0.8%) | 2 (18.2%) | 0.013 | 2 (11.8%) | 0.043 |
| Mortality (30 days) | 0 | 0 | 1.000 | 0 | 1.000 |
| Re-admission | 19 (7.3%) | 2 (18.2%) | 0.204 | 2 (11.8%) | 0.629 |

## Discussion

### Existing data

To our knowledge, there are no data combining or correlating pre-existing AC, the time point of postoperative AC, and risk factors with PH and TE in patients undergoing craniotomy for meningioma. By contrast, there are data on "early" AC, which indicate the safety of administering anticoagulant agents within 24 hours after the procedure. Unfortunately, in no studies were different time points of AC juxtaposed [16–18]. In addition, several studies can be found on patients with various diagnoses in which different AC regimens are compared, such as mechanical vs. chemical prophylaxis or chemical prophylaxis vs. placebo. The results indicate that anticoagulant agents seem to be particularly effective at preventing TE, whereas the hemorrhage rates differ from one study to the next [3, 19–29]. Although studies addressing AC in patients undergoing craniotomy for meningioma exist, they mainly focus on whether or not chemical prophylaxis should be administered [8, 30]. Similarly, we found data on prophylactic treatment in patients operated on for a high-grade glioma [31, 32], as well as data for venous thromboembolism and intracranial hemorrhage after craniotomy for primary malignant brain tumors [33] as well as prophylaxis in patients who had undergone decompressive craniectomy [34]. Wang et al. elucidated the risk and benefits of heparin usage in adult patients receiving neurosurgery in a systematic review and meta-analysis [35]. However, there are no data on specific time points or comparisons of dosage when previous AC treatment is assessed.

### Delayed anticoagulation raises the risk of thromboembolic events

Our data clearly indicate that the later AC is performed, the higher the risk of TE. This is consistent with previous findings in which neurosurgical patients showed an increased risk for symptomatic TE, especially due to immobilization and lengthy operative procedures [36, 37]. Risk factors in meningioma patients were highlighted by Nunno et al. [38]. The thromboembolism rate of 3.38% is in keeping with our data. However, rates of DVT in screening studies revealed higher rates of asymptomatic TE [39]. Pre-existing data revealed a decreased risk for patients with continuous heparin treatment [40], although the study compared a combined chemical and mechanical regimen (IPC, leg raise) with subsequent chemical treatment and stockings. Interestingly, this study found no increase in hemorrhage due to continuous heparinization. Regarding the CLOTS trial [41], our management rules did not provide for IPC, although other sources found it to be useful in a strictly neurosurgical collective in an RCT [42]. We will therefore consider the additional use of IPC in future treatment.

Multiple logistic regression was used to test for independent risk factors for TE. Besides the time point of AC, maximum white blood cell count, white blood cell count on admission, and minimum hematocrit, maximum systolic blood pressure and minimum INR were included in the analysis to obtain the model with the best possible fit.

The time point of AC retains its significance and can therefore be considered an independent risk factor for thromboembolic events (p = 0.008). The OR of 1.029 (95% CI 1.007–1.051)

shows that the TE rate increases significantly with subsequent anticoagulation. Other independent risk factors are higher white blood cell count on admission, lower maximum white blood cell count, and low minimum hematocrit. The low number of patients in both the TE and PE group do not provide a good model fit. Even so, the low p value of the time point of anticoagulation is striking. A larger number of patients is necessary for an accurate risk profile.

Two patients were diagnosed with thromboembolic events before the regular administration of heparin. Both were therapeutically anticoagulated immediately with unfractionated heparin. Neither of them suffered from PH (Table 3). On the other hand, anticoagulation was delayed in two patients with PH (96/240h post-op.) because of intracranial bleeding, who both later suffered from thromboembolism. A causal association cannot be established of course in these rare cases.

## "Early" anticoagulation does not raise the risk of intracranial hemorrhage

The American College of Chest Physicians considers any intracranial operation a procedure with increased risk of bleeding [43]. In line with recent data, we found no increase in postoperative hemorrhage after the early administration of LMWH. Moreover, neither heparin administered preoperatively [30] nor continuous perioperative administration [40] increased the risk of PH. We therefore declare that perioperative prophylactic heparinization in meningioma patients is always safe regarding intracranial hemorrhage. Furthermore, Nittby et al. found that most patients suffering from postoperative hemorrhage had not had any kind of anticoagulation in their study [44]. This is consistent with our data, in which only 3 of 17 patients with PH had been anticoagulated between surgery and PH.

In multiple logistic regression to find independent risk factors for PH, only late AC retained its significance. This is because after PH, heparin initiation is usually postponed. No other independent risk factors for PH were found.

## Outcome

There were no deaths due solely to postoperative hemorrhage. Two patients died as a direct result of pulmonary embolism: one in uroseptic shock after de-escalation of therapy at the patient's request, the other due to cranial infarction of unknown origin after surgical revision of intracranial hemorrhage. Generally speaking, PH appears to be more treatable and has better survival rates than TE. This also applies to neurological outcome on GOS and mRS, which is less severely lowered by hemorrhage.

## Conclusions

In summary, postoperative hemorrhage is mostly a complication of surgery itself or of a pre-existing medical condition, and not of chemical prophylaxis. As surgeons, we tend to exaggerate surgical complications, such as perioperative hemorrhage, because they are inextricably linked to our operative routine. Although both TE and PH do indeed affect patients' outcomes, in our study TE was found to have a far more severe effect on patients' outcomes. Previous data suggested high rates for reduced outcomes for patients with TE [39] or PH, but there was no direct comparison to post-hemorrhage outcomes. According to our findings, the definite risks of thromboembolism outweigh the harm caused by postoperative intracranial hemorrhage. This should be considered when deciding postoperative anticoagulation treatment.

To further elucidate anticoagulation management in neurosurgical patients, we therefore propose: i) a prospective randomized study with continuously administered prophylactic AC vs. current regimen, ii) integrating this study with data from different entities, such as glioma,

spontaneous and traumatic intracranial hemorrhage. We also suggest prophylactic AC on day 1 after meningioma surgery.

## Supporting information

**S1 Data.**
(XLS)

## Author Contributions

**Conceptualization:** Florian Wilhelmy, Annika Hantsche, Stefan Rasche, Dirk Lindner, Jürgen Meixensberger.

**Data curation:** Florian Wilhelmy, Annika Hantsche, Tim Wende, Clara Frydrychowicz, Dirk Lindner, Jürgen Meixensberger.

**Formal analysis:** Florian Wilhelmy, Annika Hantsche, Johannes Kasper, Dirk Lindner.

**Investigation:** Florian Wilhelmy, Annika Hantsche, Dirk Lindner.

**Methodology:** Florian Wilhelmy, Annika Hantsche, Dirk Lindner, Jürgen Meixensberger.

**Project administration:** Florian Wilhelmy, Annika Hantsche, Dirk Lindner, Jürgen Meixensberger.

**Resources:** Florian Wilhelmy, Jürgen Meixensberger.

**Supervision:** Florian Wilhelmy, Stefan Rasche, Dirk Lindner.

**Validation:** Florian Wilhelmy, Vera Reuschel, Clara Frydrychowicz, Jürgen Meixensberger.

**Visualization:** Vera Reuschel.

**Writing – original draft:** Florian Wilhelmy, Annika Hantsche.

**Writing – review & editing:** Florian Wilhelmy, Annika Hantsche, Tim Wende, Johannes Kasper, Vera Reuschel, Clara Frydrychowicz, Stefan Rasche, Dirk Lindner, Jürgen Meixensberger.

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
