## [Decision Letter · Decision Letter 0]

9 Jun 2020

PONE-D-20-13930

Perioperative Anticoagulation in Patients with Intracranial Meningioma: No increased risk of intracranial hemorrhage?

PLOS ONE

Dear Dr. Wilhelmy,

Thank you for submitting your manuscript to PLOS ONE. After careful consideration, we feel that it has merit but does not fully meet PLOS ONE’s publication criteria as it currently stands. Therefore, we invite you to submit a revised version of the manuscript that addresses the points raised during the review process.

We look forward to receiving your revised manuscript.

Kind regards,

Jonathan H Sherman

Academic Editor

PLOS ONE

Journal Requirements:

2. In ethics statement in the manuscript and in the online submission form, please provide additional information about the patient records used in your retrospective study. Specifically, please ensure that you have discussed whether all data were fully anonymized before you accessed them and/or whether the IRB or ethics committee waived the requirement for informed consent. If patients provided informed written consent to have data from their medical records used in research, please include this information.

3. Please change your reference to "p-value=NS to the exact value.

5. Please amend your authorship list in your manuscript file to include author Johannes Casper.

6. Please amend the manuscript submission data (via Edit Submission) to include author Johannes Dietterle.

7. We note you have included a table to which you do not refer in the text of your manuscript. Please ensure that you refer to Table 2, 4, 5 in your text; if accepted, production will need this reference to link the reader to the Table.

Reviewers' comments:

Reviewer's Responses to Questions

**Comments to the Author**

1. Is the manuscript technically sound, and do the data support the conclusions?

Reviewer #1: No

Reviewer #2: Yes

2. Has the statistical analysis been performed appropriately and rigorously? 

Reviewer #1: I Don't Know

Reviewer #2: Yes

3. Have the authors made all data underlying the findings in their manuscript fully available?

Reviewer #1: Yes

Reviewer #2: Yes

4. Is the manuscript presented in an intelligible fashion and written in standard English?

Reviewer #1: No

Reviewer #2: No

5. Review Comments to the Author

Reviewer #1: The authors utilize their retrospective series of patients undergoing craniotomy for meningioma to assess the perioperative use of anticoagulation on patient outcome. The decision to use anticoagulation and timing of initiating/re-initiating anticoagulation following surgery remains a controversial topic without clear consensus. The authors found that delayed initiation of anticoagulation was associated with significantly increased rate of thromboembolic events. Early initiation of did not result in increased rates of post-operative hemorrhage. Additionally, they note that thromboembolic events resulted in poor patient outcomes than post-operative hemorrhage, in their series. I would refer to the authors to the following criticisms:

1. The organization of this manuscript makes it challenging to adequately assess several important aspects of the study.

2. The institutional protocol for use and timing of anti-coagulation following surgery is not clearly presented. The reason for delaying anticoagulation and the rationale for resuming anticoagulated needs to be more clearly elaborated upon and presented in a more logical fashion. While this is attempted in the “Practice at our clinical site” portion of the discussion, this should be revised to more appropriately reside in the “Methods” section.

3. The post-operative imaging protocol is not clearly defined. Do all patients get immediate post-op CTs or at a certain time following surgery?

4. The positive findings from the regression analyses should have been more clearly explained (or attempted to be explained) in the discussion.

5. Were the 286 menigioma patients operated on at the same institution/multi-institution?

6. Table 1 – thoughts on why blood pressure was 20 points higher on systolic; does this play a role in increased hemorrhage rates? What is the timing of these blood pressures.

Reviewer #2: I think the language is slightly confusing overall and maybe a good proofreading would help make things more clear.

The introduction talks about what seems to be therapeutic anticoagulation for patients with pre existing conditions and in the abstract it seems like they are talking about perioperative prophylactic anticoagulation for DVT. Can the authors make this more clear.

It is unclear why the tables are set up as control group and the outcome they are looking for Thrombotic event or hemorrhage. They should not use the term control group as this makes it seem like they were stratified. They should just call it no event.

In the results discussing delayed anticoagulation it is unclear if this is therapeutic or prophylactic.

I think in their conclusions they discuss that this is all prophylactic dose but this is important to make clear throughout the manuscript.

I think it seems like the focus is on prophylactic doses the introduction could be better written to reflect data on chemical prophylaxis and make the manuscript more clear that is the focus

6. PLOS authors have the option to publish the peer review history of their article (what does this mean?). If published, this will include your full peer review and any attached files.

Reviewer #1: No

Reviewer #2: No

---

## [Author Response · Author response to Decision Letter 0]

23 Jul 2020

Comments to the Author

1. Is the manuscript technically sound, and do the data support the conclusions?

Reviewer #1: No

Reviewer #2: Yes

2. Has the statistical analysis been performed appropriately and rigorously? 

Reviewer #1: I Don't Know

Reviewer #2: Yes

 Multiple Regression has been further discussed (see below).

3. Have the authors made all data underlying the findings in their manuscript fully available?

Reviewer #1: Yes

Reviewer #2: Yes

4. Is the manuscript presented in an intelligible fashion and written in standard English?

Reviewer #1: No

Reviewer #2: No

As non native speakers, our abilities are limited. In that regard, we sent the manuscript to a native specializing in revision of scientific literature and we hope to have made the manuscript intelligible.

Review Comments to the Author

Reviewer #1: The authors utilize their retrospective series of patients undergoing craniotomy for meningioma to assess the perioperative use of anticoagulation on patient outcome. The decision to use anticoagulation and timing of initiating/re-initiating anticoagulation following surgery remains a controversial topic without clear consensus. The authors found that delayed initiation of anticoagulation was associated with significantly increased rate of thromboembolic events. Early initiation of did not result in increased rates of post-operative hemorrhage. Additionally, they note that thromboembolic events resulted in poor patient outcomes than post-operative hemorrhage, in their series. I would refer to the authors to the following criticisms:

1. The organization of this manuscript makes it challenging to adequately assess several important aspects of the study.

We revised the “Methods” section and placed the therapy regimen and interdisciplinary course of treatment here. We hope that this further elucidates the need for standard protocols, as we lack them. Furthermore we hope to make decision making at our site more clear. We are aware that the variety of endpoints and the organization of the manuscript are challenging. We tried to address this in our revised discussion and to focus on the important conclusions.

2. The institutional protocol for use and timing of anti-coagulation following surgery is not clearly presented. The reason for delaying anticoagulation and the rationale for resuming anticoagulated needs to be more clearly elaborated upon and presented in a more logical fashion. While this is attempted in the “Practice at our clinical site” portion of the discussion, this should be revised to more appropriately reside in the “Methods” section.

Unfortunately, the lack of any institutional protocol sparked this investigation. We tried to further elucidate the decision making process regarding AC and put it in the “Methods” section, were it absolutely should reside. Again, we have difficulties to point out a clear concept or algorithm. nonetheless the variety of different therapy regimen allowed us to investigate the influence of different timepoints of AC. The section added reads as follows:

Patients were anticoagulated based on national guidelines [14, 15], known independent risk factors and preexistent diseases. Time-point and dose as well as the choice of substance were chosen according to hospital’s guidelines and in interdisciplinary bed-side discourse with intensive care and neurosurgical practitioners. However, anticoagulation in this study did not follow a definite protocol, which provides the variety of timepoints and therapy regimen that we were able to examine. The common course of treatment and diagnostic can be described as follows:

Patients are not anticoagulated until a CT scan has been performed. CT scans on the first postoperative day are routine (within 24h after surgery). Upon new neurological deficits CT-scans are performed immediately.

We began prophylactic AC treatment in patients undergoing intracranial surgery on the first day after surgery, as far as postoperative CT-scans do not show any signs of residual bleeding. Patients with postoperative or perioperative hemorrhage are mainly anticoagulated after 3-5 days, after a repetitive CT-scan without any sign of a growing hemorrhage.

In patients with preoperative regimen of DOAC or NOAC, an interruption of therapy up to 3 weeks was carried out, dependent on indication, during the examined period of time. In those cases, heparin-bridging was regularly undertaken.

Since 2014, a standard operating procedure states that hospitalized tumor patients should be considered for anticoagulation as long as there are no contraindications, in which case mechanical prophylaxis or no prophylaxis at all should be applied. We adapted the regimen and are currently halting preexisting AC until suture’s removal, if no urgent indication exists.

3. The post-operative imaging protocol is not clearly defined. Do all patients get immediate post-op CTs or at a certain time following surgery?

Patients are routinely scanned on the day following surgery. This is usually done “the next morning” between 8am and 12am, depending on ICUs capacity to oversee CT-transport. We overlooked the data and CT scans are exactly tracked with timestamps in our data. From that we can conclude, that they were all undertaken within 24h after surgery (at least). In case of new neurological deficits, CT-scans were performed immediately as emergency diagnostic. In brief we summarized for the manuscript: 

Patients are not anticoagulated until a CT scan has been performed. CT scans on the first postoperative day are routine (within 24h after surgery). Upon new neurological deficits CT-scans are performed immediately.

4. The positive findings from the regression analyses should have been more clearly explained (or attempted to be explained) in the discussion.

Absolutley. The most important positive finding is th persistence of late AC timepoints as independent risk factor, in our opinion. Therefore we added the following sections:

In: “Delayed anticoagulation increases risk of thromboembolic events”

Multiple logistic regression was used to test for independent risk factors for TE. Besides timepoint of AC, maximum white blood cell count, white blood cell count at admission, minimum hematocrit, maximum systolic blood pressure and minimum Quick – value entered the analysis to achieve the model with the best possible fit.

Timepoint of AC remains its significance and can therefore be considered an independent risk factor for thromboembolic events (p=0.008). The Odd’s Ratio of 1.029 (95% CI 1.007-1.051) displays that the TE rate increases significantly with later anticoagulation. Other independent risk factors are higher white blood cell count at admission, lower maximal white blood cell count (?) and low minimal hematocrit.

The low number of patients in both, the TEE and PE group, do not make for a good model fit. Still, the low p-value of the timepoint of anticoagulation is striking. But a larger number of patients is necessary for an accurate risk profile.

And in: “Early” anticoagulation does not increase the risk of intracranial hemorrhage

In multiple logistic regression to find independent risk factors for PH, only late AC retained its significance. This is because after PH heparin initiation is usually postponed. Other independent risk factors for PH could not be found.

5. Were the 286 menigioma patients operated on at the same institution/multi-institution?

All in the same institution. We added the hospital in the following passage: 

The digital database of the University Hospital Leipzig for all patients with the histopathological diagnosis of meningioma, who underwent surgery in 2012-2018 and operated in the Department of Neurosurgery, University Hospital Leipzig, was searched.

6. Table 1 – thoughts on why blood pressure was 20 points higher on systolic; does this play a role in increased hemorrhage rates? What is the timing of these blood pressures.

Yes a very interesting finding, which we spent a lot of time investigating, actually. We first correlated high blood pressure and PH rates, then saw it prevail in multiple regression. We then correlated the time of blood pressure peaks to 1.) time point of bleeding and blood pressure peak and time of bleeding and 2.) time point of blood pressure peak after surgery (we performed the same for hematocrit, pTT, INR etc.). The result was chaotic, yet clear: There is no association whatsoever between WHEN the peak happens and WHEN the bleeding happens. We then assessed blood pressure ad admission, to also find a (weaker) correlation. We therefor concluded: The disposition for high blood pressure in general (due to comorbidities) is a risk factor for rebleeding, regardless of certain “peaks”. We know the data is insufficient and we would need 1.) continuous blood pressure measurement data, 2.) blood-pressure-curve analyses a.s.o. We did not include this in the paper due to the large amount of data and figures it would need to explain but we have the topic and mind and will further investigate and, if the data is consistent, publish separately. For our objective of AC we found it not be too relevant.

Reviewer #2:

1. I think the language is slightly confusing overall and maybe a good proofreading would help make things more clear.

As non native speakers, our abilities are limited. In that regard, we sent the manuscript to a native specializing in revision of scientific literature and we hope to have made the manuscript intelligible.

2. The introduction talks about what seems to be therapeutic anticoagulation for patients with pre existing conditions and in the abstract it seems like they are talking about perioperative prophylactic anticoagulation for DVT. Can the authors make this more clear.

We revised the very first sentence to mention AC as follows: 

In perioperative treatment of patients undergoing intracranial surgery we face opposed poles: The necessity for prophylactic AC and its most feared adverse effect, intracranial hemorrhage.

We hope to set the goal of this study right at the beginning! Secondly we added the following passage, to make clear, that preexisting AC is not examined, but is an additional risk factor and underlines the necessity of proper prophylactic AC:

Neurosurgical patients are at high risk for thromboembolism of any kind due to immobilization and length of surgery. Additionally, Ddue to demographic changes, making cardiovascular diseases more prevalent, increasing numbers of significantly anticoagulated patients are observed [4]. A survey conducted by Skardelly et al. concluded that in 2016 nearly two thirds of the German neurosurgical departments had not yet defined an algorithm regarding continuous, discontinuous or bridged anticoagulant and antiplatelet therapy during elective surgery.

It is unclear why the tables are set up as control group and the outcome they are looking for Thrombotic event or hemorrhage. They should not use the term control group as this makes it seem like they were stratified. They should just call it no event.

Absolutely, ther was no prospective investigation and no negative control. We called the group “no event” and used the abbreviation “NO”.

In the results discussing delayed anticoagulation it is unclear if this is therapeutic or prophylactic.

We clarified in the text:

Time - point of Prophylactic Anticoagulation

Patients with delayed prophylactic AC (point of treatment in hours after surgery) were more likely to suffer from TE (CGNO 29 ± 21.9h, TE 54 ± 69.2, p=0.301).

The switch to therapeutic anticoagulation in 2 cases of early postoperative pulmonary embolism is indeed statistically problematic. We tried to address this in the section below the passage mentioned.

I think in their conclusions they discuss that this is all prophylactic dose but this is important to make clear throughout the manuscript.

Asolutely! We added the “prophylactic” in all places were it needed clarification.

I think it seems like the focus is on prophylactic doses the introduction could be better written to reflect data on chemical prophylaxis and make the manuscript more clear that is the focus

 Now that you point it out, we agree clearly. We differentiated between prophylactic when and wherever necessary.

---

## [Decision Letter · Decision Letter 1]

17 Aug 2020

Perioperative Anticoagulation in Patients with Intracranial Meningioma: No increased risk of intracranial hemorrhage?

PONE-D-20-13930R1

Dear Dr. Wilhelmy,

We’re pleased to inform you that your manuscript has been judged scientifically suitable for publication and will be formally accepted for publication once it meets all outstanding technical requirements.

Kind regards,

Jonathan H Sherman

Academic Editor

PLOS ONE

Additional Editor Comments (optional):

Reviewers' comments:

Reviewer's Responses to Questions

**Comments to the Author**

1. If the authors have adequately addressed your comments raised in a previous round of review and you feel that this manuscript is now acceptable for publication, you may indicate that here to bypass the “Comments to the Author” section, enter your conflict of interest statement in the “Confidential to Editor” section, and submit your "Accept" recommendation.

Reviewer #1: All comments have been addressed

Reviewer #2: All comments have been addressed

2. Is the manuscript technically sound, and do the data support the conclusions?

Reviewer #1: Yes

Reviewer #2: Yes

3. Has the statistical analysis been performed appropriately and rigorously? 

Reviewer #1: Yes

Reviewer #2: I Don't Know

4. Have the authors made all data underlying the findings in their manuscript fully available?

Reviewer #1: Yes

Reviewer #2: Yes

5. Is the manuscript presented in an intelligible fashion and written in standard English?

Reviewer #1: Yes

Reviewer #2: Yes

6. Review Comments to the Author

Reviewer #1: The authors have addressed the concerns that were brought up in the revision requests. Satisfied with the responses.

Reviewer #2: The authors have addressed all the comments adequately I have no issue with this being published.

7. PLOS authors have the option to publish the peer review history of their article (what does this mean?). If published, this will include your full peer review and any attached files.

Reviewer #1: **Yes: **Walavan Sivakumar

Reviewer #2: No

---

## [Editor Report · Acceptance letter]

19 Aug 2020

PONE-D-20-13930R1 

Perioperative Anticoagulation in Patients with Intracranial Meningioma: No increased risk of intracranial hemorrhage? 

Dear Dr. Wilhelmy:

I'm pleased to inform you that your manuscript has been deemed suitable for publication in PLOS ONE. Congratulations! Your manuscript is now with our production department. 

Kind regards, 

on behalf of

Dr. Jonathan H Sherman 

Academic Editor

PLOS ONE